# No-Regret Learning of Nash Equilibrium
# for Black-Box Games via Gaussian Processes

**Minbiao Han**[*1]          **Fengxue Zhang** [*1]          **Yuxin Chen**[1]

[1]Department of Computer Science, University of Chicago, Chicago, Illinois, USA

## Abstract

This paper investigates the challenge of learning in black-box games, where the underlying utility function is unknown to any of the agents. While there is an extensive body of literature on the theoretical analysis of algorithms for computing the Nash equilibrium with *complete information* about the game, studies on Nash equilibrium in *black-box* games are less common. In this paper, we focus on learning the Nash equilibrium when the only available information about an agent's payoff comes in the form of empirical queries. We provide a no-regret learning algorithm that utilizes Gaussian processes to identify the equilibrium in such games. Our approach not only ensures a theoretical convergence rate but also demonstrates effectiveness across a variety collection of games through experimental validation.

## 1 INTRODUCTION

The Nash equilibrium (NE) is a fundamental concept in game theory and represents a stable point in strategic interactions among multi-agent systems. The computation of NE has been extensively explored. Existing computational studies [Basar, 1987, Li and Basar, 1987, URYAs' Ev and Rubinstein, 1994] have provided valuable insights into equilibrium existence, complexity, and algorithmic solutions when agents' utility information is public knowledge. However, when dealing with a game, particularly one involving multiple agents, it is unrealistic to expect that anyone possesses an explicit representation of its utility function, even if the game itself has a succinct representation. In many real-world scenarios, a reasonable modeling assumption is

---
[*]Equal Contribution, the author names are in alphabetical order.

that given the strategy profile of all agents, we can query their corresponding utilities.

Our focus lies in developing algorithms that discover NE through a series of queries, where each query proposes a strategy profile and receives information about the corresponding utilities of all agents. Such games are also referred to as black-box or simulation-based games [Wellman, 2006, Jordan et al., 2008, Vorobeychik, 2010, Fearnley et al., 2015]. For instance, we can envision an agent-based combat simulation where the analyst has the ability to configure the strategic parameters of the adversaries and execute the simulation to obtain a representative outcome of a battle or campaign [Vorobeychik and Porche, 2009]. Other examples include simulation-based game theoretic analyses of supply chains [Vorobeychik et al., 2006] and simultaneous ascending auctions [Wellman et al., 2008]. The motivation of this model is from a common practice today of "*centralized training, decentralized execution*" in multi-agent learning (originated from the highly impactful work of Lowe et al. [2017]). That is, in many robotics and game-playing applications (e.g., OpenAI Gym), the learning environments are well-defined such that the game parameters can be learned in a centralized fashion by controlling agents' action profiles. Thus, the agents can learn to play the NE strategy from the perspective of a centralized game analyst, and then deploy the learned strategies in the decentralized environment to play against unknown opponents.

In order to learn the NE of the aforementioned black-box games through queries, it is crucial to estimate the distance of each query from the NE. Essentially, we can estimate whether each agent has an inclination to deviate from the queried strategy. As a result, each query involves computing the optimal deviation of all agents from the specified strategy. This process is inherently computationally expensive, as it requires optimization of an unknown utility function for each agent. To summarize, we make the following assumption about the agents' utility function in the black-box games mentioned above.

**Assumption 1.** *We assume the utility functions may have some regularity properties but are possibly strongly non-convex. Queries on the utility functions result from an expensive process and can be corrupted by noise.*

In light of the above assumption and the intrinsic cost of querying utility functions, we employ Gaussian Process (GP) [Garnett, 2023] as an effective tool for tackling such black-box optimization problems. This paper investigates the application of GP in the context of learning the Nash equilibrium.

**Our Results and Implications.** Given the lack of agents' utility information and the expensive query mentioned above, this paper studies efficient no-regret learning of the NE for black-box games via GP. To the best of our knowledge, there were no existing GP algorithms for learning NE with a known no-regret guarantee. The key innovation in our work is the design of a novel GP objective specifically for NE learning. Specifically, we characterize the equilibrium computation as an optimization problem involving an unknown loss function. This function represents the maximum utility gain that agents can achieve by deviating from the given strategy. Notably, reaching a zero value of this function corresponds to the NE, a scenario where no agent can improve their utility by changing their strategy given the strategies of others.

A critical aspect of our approach is that each query to the loss function involves calculating all agents' optimal deviation from the given strategy. This process is inherently computationally expensive, as it requires optimization of an unknown utility function for each agent. Our main result provides a no-regret learning algorithm that provides a theoretical guarantee of convergence to the Nash Equilibrium. We demonstrate the algorithm's effectiveness and compare its performance in terms of regret against recent algorithms in the literature on a collection of classical structured games as well as the real-world marketing budget allocation game.

## 2 RELATED WORK

### 2.1 BAYESIAN OPTIMIZATION APPROACH

The most closely related line of research focuses on addressing game-theoretic models that are computationally expensive to evaluate using Bayesian Optimization (BO) techniques. Al-Dujaili et al. [2018] proposed a method to find equilibria for such games in a sequential decision-making framework using BO. Specifically, they introduced the *game-theoretical regret* of a strategy profile $x$ as the most utility any agent $i$ can gain by deviating from $x_i$ to any strategy in $\mathcal{X}_i$. The authors employ BO to minimize an approximation of the game-theoretical regret and approximate the pure strategy NE. The performance in terms of game-

theoretical regret of the proposed method is validated on a collection of synthetic games by comparison with some recent algorithms.

Picheny et al. [2019] also studied the same problem of solving games with the GP-based approach. The main difference between this paper and Al-Dujaili et al. [2018] is the acquisition function used by BO. Instead of minimizing the game-theoretical regret like Al-Dujaili et al. [2018], Picheny et al. [2019] proposed two acquisition functions. Specifically, one acquisition function is to maximize the probability of achieving the equilibrium, while the other one is to reduce as quickly as possible an uncertainty measure related to the equilibrium.

Marchesi et al. [2020] proposed a multi-arm bandit algorithm on top of the Gaussian processes and offers theoretical justification. Our work differentiates from two perspectives. First, Marchesi et al. [2020] focused on two-player zero-sum games, while our work allows multi-player normal-form games. Second, the regret analysis in Marchesi et al. [2020] relied on a suboptimal gap in the denominators of the regret bound. As discussed by Lattimore and Szepesvári [2020], the major problem with this dependency is that this gap, in practice, could be arbitrarily small and downgrade the practicality of the resulting regret analysis. At the same time, our theoretical results of the regret bound rely on the maximum mutual information of GP instead and are gap-independent.

Recently, Aprem and Roberts [2021] studied a specific form of games, termed potential games [Monderer and Shapley, 1996]. Specifically, they utilized the structure of potential games and proposed to use a Gaussian process model for the potential function directly instead of modeling the utility functions like Picheny et al. [2019].

Compared to the previous work, the key contribution of our work is that we have a novel GP objective for NE learning. Furthermore, we present a no-regret learning algorithm that guarantees convergence to NE, addressing a gap in the existing literature, which lacked theoretical convergence analysis for similar approaches.

### 2.2 OTHER ONLINE LEARNING ALGORITHMS

Learning Nash Equilibria has been widely studied in the literature. Regret minimization serves as a closely related category of learning rules. In essence, an agent incurs ex-post regret if, during certain periods, they could have achieved a higher average payoff by choosing a different strategy. Several straightforward learning procedures exist that aim to minimize ex-post regret [Foster and Vohra, 1999, Hart and Mas-Colell, 2000, 2001, Sessa et al., 2019]. However, it is important to note that relying on ex-post regret minimization rules does not guarantee behaviors consistently converging to the Nash equilibrium. What the evidence supports is that these rules cause the empirical frequency distribution of play

to converge to the set of correlated equilibria, which, while including Nash equilibria, is frequently much larger and not necessarily more desirable in terms of strategic outcomes.

Another relevant learning rule is regret testing [Foster and Young, 2006]. Here, an agent compares their average per-period payoff over an extended sequence of plays with the average obtained through occasional experiments with alternative strategies. Foster and Young [2006] demonstrated that, for all finite two-person games, this rule approximates Nash equilibrium behavior most of the time. Moreover, Germano and Lugosi [2014] later established that a modification of this procedure comes close to Nash equilibrium behavior in any finite $n$-person game with generic payoffs.

Another, less closely related, set of learning rules is those based on interactive learning by trials [Karandikar et al., 1998, Young, 2009, Marden et al., 2009]. In this context, an agent learns through trial and error by occasionally experimenting with new strategies, and discarding choices that fail to yield higher payoffs. They demonstrate the ability to approach pure Nash equilibrium and play a high proportion of the learning period, but typically they do not converge.

Recently, Gemp et al. [2024] proposed a novel loss function for Nash equilibrium learning in general games that is amenable to Monte Carlo estimation and allows applying SGD for efficient optimization. Though tackling a similar problem from different perspectives, the combination of a gradient-based optimizer with a Monte-Carlo estimator and a GP-based bandit algorithm has drawn interest in BO literature [Balandat et al., 2020] and indicates an interesting future direction.

Similar to the work by Aprem and Roberts [2021], Chapman et al. [2013] also studied convergence to Nash equilibria in potential games with rewards that are initially unknown. Different from the Bayesian optimization approach, they proposed a multi-agent version of Q-learning to estimate the reward functions using novel forms of the $\epsilon$–greedy learning policy. Jordan [1991] studied Bayesian learning of equilibrium, assuming each agent knows their utility information but not others. This work is also related to learning other equilibrium concepts in game theory and Bayesian optimization with multiple structured utility functions, we refer to Appendix A for more detailed discussions and comparisons.

## 3 PRELIMINARIES AND PROBLEM SETUP

We consider the optimization problem of finding the equilibrium $\boldsymbol{x}^* \in \mathcal{X}$ [1] of a game played by multiple agents, defined

as follows

$$x_i^* \in \arg \max_{x_i \in \mathcal{X}_i} u_i(x_i, \boldsymbol{x}_{-i}^*), \quad \forall i \in [n] \qquad (1)$$

where $[n] = \{1, \cdots, n\}$ denotes the set of agents, $\mathcal{X}_i$ is the action set of agent $i$ ($\mathcal{X} = \mathcal{X}_1 \times \cdots \times \mathcal{X}_n$), and $u_i(x_i, \boldsymbol{x}_{-i}^*)$ is agent $i$'s utility function where $x_i$ represents agent $i$'s action and $\boldsymbol{x}_{-i}^*$ denotes all the other agents' actions except for $i$. Our paper specifically focuses on finite games, which involve a finite number of players and a finite number of actions for each player. It is well-established, as demonstrated by Nash [1950], that every finite game possesses at least one Nash equilibrium, commonly known as the Nash existence theorem.

The problem setup is a repeated game among $N$ agents or players. Each agent $i$ has an action set $\mathcal{X}_i \subseteq \mathbb{R}^{d_i}$ and a utility function $u_i : \mathcal{X} = \mathcal{X}_1 \times \cdots \mathcal{X}_n \to [0, 1]$. We denote *all* agents' action $\boldsymbol{x} = (x_1, \cdots, x_n)$ as an action profile. The Nash Equilibrium (NE) $\boldsymbol{x}^* \in \mathcal{X}$ is denoted in Equation (1). Given any action profile $\boldsymbol{x}$, we denote a loss function $f : \mathcal{X} \to \mathbb{R}$ as follows.

$$f(\boldsymbol{x}) = \sum_{i \in [n]} \max_{x_i' \in \mathcal{X}_i} u_i(x_i', \boldsymbol{x}_{-i}) - u_i(\boldsymbol{x}) \qquad (2)$$

Note that $f(\boldsymbol{x}) \geq 0$ for all $\boldsymbol{x} \in \mathcal{X}$ and the NE $\boldsymbol{x}^* = \arg \min_{\boldsymbol{x} \in \mathcal{X}} f(\boldsymbol{x})$ satisfies $f(\boldsymbol{x}^*) = 0$. An approximate Nash equilibrium $\boldsymbol{x}$ is denoted as $\epsilon$-*NE* [Tijs, 1981, Lipton et al., 2003], where each agent's strategy, given other agents' strategies, has suboptimality at most $\epsilon$, i.e., $\max_{x_i' \in \mathcal{X}_i} u_i(x_i', \boldsymbol{x}_{-i}) - u_i(\boldsymbol{x}) \leq \epsilon, \forall i \in [n]$.

**Example 1.** *We consider a two-player game from Al-Dujaili et al. [2018], Paruchuri et al. [2008] as a running example, where the utility functions of the two players are defined as $u_1(x_1, x_2) = (x_2 - x_2^*)^2 - (x_1 - x_1^*)^2$ and $u_2(x_1, x_2) = (x_1 - x_1^*)^2 - (x_2 - x_2^*)^2$. $\boldsymbol{x}^* = (x_1^*, x_2^*) = (0.5, 0.5)$ denotes the NE. We illustrate the agent's utility function and loss function Equation (2) in Figure 1.*

Our objective is to minimize the unknown function (Equation (2)), given only the query access to the objective function. Specifically, at every time step $t$, we can query an action profile $\boldsymbol{x}^t$ and observe each agent's corresponding utility $\boldsymbol{y}^t$, where $y_i^t = u_i(\boldsymbol{x}^t) + \epsilon_i$ and $\epsilon_i \sim \mathcal{N}(0, \sigma^2)$. We denote a sequence of function evaluations (FEs) as $\mathcal{D}^{1:t} = \{(\boldsymbol{x}^1, \boldsymbol{y}^1); \cdots; (\boldsymbol{x}^t, \boldsymbol{y}^t)\}$. We define

$$f(\boldsymbol{x}^t) - f(\boldsymbol{x}^*) = f(\boldsymbol{x}^t) \qquad (3)$$

as regret, since $f(\boldsymbol{x}^*) = 0$ for NE. We want to achieve a no-regret learning of NE:

$$\lim_{T \to \infty} \frac{1}{T} \sum_t^T f(\boldsymbol{x}^t) \to 0$$

---

[1] In this paper, multiple agents' strategies are denoted by bold lowercase letters, e.g., $\boldsymbol{x}$ or $\boldsymbol{x}_{-i}$. The $i^{th}$ agent's strategy is denoted in subscript $x_i$ (non-bold).

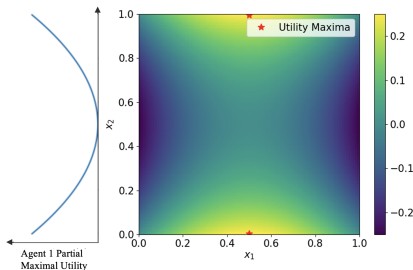 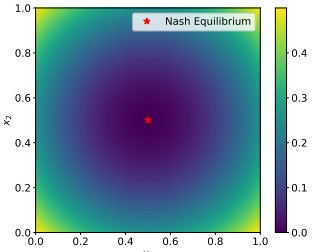 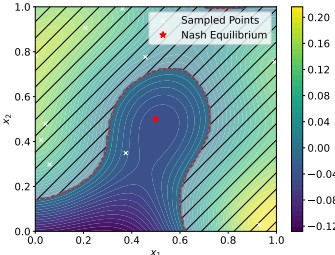

(a) Agent 1's utility in Example 1. The left plot represents agent 1's partial maximum utility from Equation (2) given agent 2's strategy $x_2$.

(b) Heatmap showing the loss function Equation (2) of Example 1. The optimal loss of 0 is attained at the NE (0.5, 0.5).

(c) LCB on Example 1's loss function posterior with 10 initialization points. Unmasked area indicates ROI defined by Equation (11).

Figure 1: Function visualizations of Example 1, where $x$-axis (i.e., $x_1$) represents agent 1's action and $y$-axis (i.e., $x_2$) represents agent 2's action. Agent 2's utility information is symmetric to Figure 1a and is therefore omitted from this plot. Figure 1a shows that a rational agent's utility maximization strategy (i.e., Utility Maxima) is highly different from the minima of the loss function (i.e., NE $(0.5, 0.5)$), which highlights the novelty and difficulty of optimizing our loss function (Equation (2)). Figure 1c highlights the efficiency of our optimization algorithm by reducing the search space.

The definition of no-regret learning of Nash equilibrium generalizes the no-regret notion in games discussed by Jafari et al. [2001], Daskalakis et al. [2021], and resembles the common notion of no-regret in the Bayesian optimization literature [Srinivas et al., 2009, Chowdhury and Gopalan, 2017]. For every agent $i \in [n]$, we model their utility function $u_i : \mathcal{X} \to [0, 1]$ as a GP, which is a probability distribution over functions, i.e.

$$u_i(\boldsymbol{x}) \sim \mathcal{GP}\big(\mu_{u_i}(\cdot), k_{u_i}(\cdot, \cdot)\big),$$

specified by its mean $\mu_{u_i}(\cdot)$ and covariance (or kernel) $k_{u_i}(\cdot, \cdot)$, respectively. The corresponding hyper-parameters are denoted by $\theta_{u_i}$. We assume every agent has the same GP prior $\mathcal{GP}(0, k(\boldsymbol{x}, \boldsymbol{x}'))$ for their utility function. Given a history of observations $\mathcal{D}^{1:t}$, the posterior distribution under a $\mathcal{GP}(0, k(\boldsymbol{x}, \boldsymbol{x}'))$ prior is also Gaussian, with mean and variance functions updated as follows.

$$\mu_{u_i,t}(\boldsymbol{x}) = \mathbf{k}_{u_i}^t(\boldsymbol{x})^\top (\mathbf{K}_{u_i}^t + \sigma^2 \mathbf{I})^{-1} \boldsymbol{y}_i^{1:t}$$
$$\sigma_{u_i,t}(\boldsymbol{x})^2 = k_{u_i}(\boldsymbol{x}, \boldsymbol{x}) - \mathbf{k}_{u_i}^t(\boldsymbol{x})^\top (\mathbf{K}_{u_i}^t + \sigma^2 \mathbf{I})^{-1} \mathbf{k}_{u_i}^t(\boldsymbol{x}) \tag{4}$$

where $\mathbf{k}_{u_i}^t(\boldsymbol{x}) = [k_{u_i}(\boldsymbol{x}^j, \boldsymbol{x})]_{j \in [t]}$, $\boldsymbol{y}_i^{1:t} = [y_i^1, \cdots, y_i^t]$, and $\mathbf{K}_{u_i}^t = [k_{u_i}(\boldsymbol{x}^i, \boldsymbol{x}^j)]_{i \in [t], j \in [t]}$ is the kernel matrix.

## 4 ALGORITHMS

### 4.1 APPROXIMATION OF THE PARTIAL MAXIMUM

Before discussing the proposed algorithm, we describe the method used in [Al-Dujaili et al., 2018]. Recall that computing the loss $f(\boldsymbol{x})$ requires the values of $u_i(\boldsymbol{x})$ and

$\max_{x_i'} u_i(x_i', \boldsymbol{x}_{-i})$ (i.e. $\max_{x_i'} u_i(x_i', \boldsymbol{x}_{-i}) + u_i(\boldsymbol{x})$) for every agent $i \in [n]$. First of all, they proposed to approximate $u_i(\boldsymbol{x})$ with the mean of the GP posterior, i.e. $\mu_{u_i,t}(\boldsymbol{x})$, as denoted in Equation (4).

The more intriguing part is to approximate the *partial maximum,* i.e., $v_i(\boldsymbol{x}_{-i}) \triangleq \max_{x_i'} u_i(x_i', \boldsymbol{x}_{-i})$. As a result, its maximum can be recovered by its mean and standard deviation, i.e.,

$$\max_{x_i'} u_i(x_i', \boldsymbol{x}_{-i}) = \mu_{v_i}(x_i) + \tau \sigma_{v_i}(x_i),$$

where $\mu_{v_i}(x_i), \sigma_{v_i}(x_i)$ denote the mean and standard deviation of $v_i(\boldsymbol{x}_{-i})$, $\tau$ is a hyper-parameter of the algorithm. Formally, given the observation history $\mathcal{D}^{1:t}$, they can be computed as follows.

$$\mu_{v_i,t}(x_i) = \mathbb{E}_{x_i'}\big[\mu_{v_i,t}(x_i')\big]$$
$$\sigma_{v_i,t}^2(x_i) = \mathbb{E}_{x_i'}\big[\big(\mu_{v_i,t}(x_i') - \mu_{v_i,t}(x_i)\big)^2\big] \tag{5}$$

The function value can therefore be approximated as

$$\hat{f}(\boldsymbol{x}|\mathcal{D}^{1:t}) \approx \max_i \mu_{v_i,t}(x_i) + \tau \sigma_{v_i,t}(x_i) - \mu_{u_i,t}(\boldsymbol{x}). \tag{6}$$

Al-Dujaili et al. [2018] used Equation (6) as the acquisition function and searching the query point $\boldsymbol{x}^{t+1} = \arg\min_{\boldsymbol{x}} \hat{f}(\boldsymbol{x}|\mathcal{D}^{1:t})$ for the next round $t + 1$. However, the acquisition function in the BO should balance between exploration and exploitation in general, while maximizing Equation (6) is pure exploitation, i.e., sampling from potentially optimal areas in $\mathcal{X}$ according to the posterior of the GP model.

## 4.2 ADAPTIVE LEVEL-SET ESTIMATION FOR GLOBAL OPTIMIZATION

We take inspiration from recent advancements in high-dimensional Bayesian optimization (HDBO) by [Zhang et al., 2023] and integrate the idea of [Al-Dujaili et al., 2018] into its framework to achieve efficient optimization of the objective defined in Equation (2) with a rigorous theoretical guarantee on the convergence rate. First, We approximate the unknown $v_i(\boldsymbol{x}_{-i}) \triangleq \max_{x'_i} u_i(x'_i, \boldsymbol{x}_{-i})$ with its corresponding upper confidence bound (UCB) and lower confidence bound (LCB) derived from the marginalized $\mathcal{GP}_{v_i} \triangleq \mathcal{GP}_{u_i|\boldsymbol{x}_{-i}}$ and

$$\text{UCB}_{v_i,t}(\boldsymbol{x}_{-i}, \mathcal{S}) \triangleq$$
$$\max_{x'_i:(x'_i, \boldsymbol{x}_{-i}) \in \mathcal{S}} \mu_{u_i,t-1}(x'_i, \boldsymbol{x}_{-i}) + \beta^{1/2} \sigma_{u_i,t-1}(x'_i, \boldsymbol{x}_{-i}), \quad (7)$$

$$\text{LCB}_{v_i,t}(\boldsymbol{x}_{-i}, \mathcal{S}) \triangleq$$
$$\max_{x'_i:(x'_i, \boldsymbol{x}_{-i}) \in \mathcal{S}} \mu_{u_i,t-1}(x'_i, \boldsymbol{x}_{-i}) - \beta^{1/2} \sigma_{u_i,t-1}(x'_i, \boldsymbol{x}_{-i}), \quad (8)$$

where $\beta$ controls the confidence level and will be discussed in the later analysis. $\mathcal{S}$ denotes the domain where the marginal maximum is taken. We will show that Equation (7) and Equation (8) provide a high confidence bound of $v_i$ with its width bounded after a certain amount of iterations.

Second, we modify the superlevel-set estimation and filtering in Zhang et al. [2023] to achieve efficient search space filtering for optimization.

The original HDBO algorithm proposed by [Zhang et al., 2023], leverages the confidence interval of the global Gaussian process $\mathcal{GP}$ to define the upper confidence bound $\text{UCB}_t(\boldsymbol{x}) \triangleq \mu_{t-1}(\boldsymbol{x}) + \beta^{1/2} \sigma_{t-1}(\boldsymbol{x})$ and lower confidence bound $\text{LCB}_t(\boldsymbol{x}) \triangleq \mu_{t-1}(\boldsymbol{x}) - \beta^{1/2} \sigma_{t-1}(\boldsymbol{x})$, where $\sigma_{t-1}(\boldsymbol{x}) = k_{t-1}(\boldsymbol{x}, \boldsymbol{x})^{1/2}$ and $\beta$ acts as an scaling factor. Then the maximum of the global lower confidence bound $\text{LCB}_{t,\max} \triangleq \max_{\boldsymbol{x} \in \mathcal{X}} \text{LCB}_t(\boldsymbol{x})$ is used as the threshold for filtering the candidates with low UCB. Therefore, it defines the superlevel-set on the search space $\mathcal{X}$ that w.h.p. contains the global optimum.

Here we use the confidence interval of the global Gaussian process $\mathcal{GP}_{u_i}$ and the marginalized UCB defined in Equation (7) to define the upper confidence bound of the objective defined in Equation (2) similarly.

For each utility function $u_i$, at a certain time $t$ we have the corresponding upper and lower confidence bound:

$$\text{UCB}_{u_i,t}(\boldsymbol{x}) \triangleq \mu_{u_i,t-1}(\boldsymbol{x}) + \beta^{1/2} \sigma_{u_i,t-1}(\boldsymbol{x})$$
$$\text{LCB}_{u_i,t}(\boldsymbol{x}) \triangleq \mu_{u_i,t-1}(\boldsymbol{x}) - \beta^{1/2} \sigma_{u_i,t-1}(\boldsymbol{x}).$$

Then we have the UCB and LCB for $f$:

$$\text{UCB}_{f,t}(\boldsymbol{x}, \mathcal{S}) \triangleq \sum_{i \in [n]} \text{UCB}_{v_i,t}(\boldsymbol{x}_{-i}, \mathcal{S}) - \text{LCB}_{u_i,t}(\boldsymbol{x}) \quad (9)$$

$$\text{LCB}_{f,t}(\boldsymbol{x}, \mathcal{S}) \triangleq \sum_{i \in [n]} \text{LCB}_{v_i,t}(\boldsymbol{x}_{-i}, \mathcal{S}) - \text{UCB}_{u_i,t}(\boldsymbol{x}) \quad (10)$$

Since $f(\boldsymbol{x}^*) = 0$ means Nash Equilibrium is achieved at $\boldsymbol{x}^*$, the minimum of $\text{LCB}_{f,t}$ over a search space containing the global optimum should be smaller than $f(\boldsymbol{x}^*) = 0$ with high probability. And as $t$ approaches $\infty$, $\text{LCB}_{f,t} \to 0$. Such property will be reflected in Theorem 1 discussed below. For briefness, we ignore the $\mathcal{S}$ on the inputs when we feed $\mathcal{X}$. Namely we denote $\text{UCB}_{f,t}(\boldsymbol{x}, \mathcal{X})$ with $\text{UCB}_{f,t}(\boldsymbol{x})$, and denote $\text{LCB}_{f,t}(\boldsymbol{x}, \mathcal{X})$ with $\text{LCB}_{f,t}(\boldsymbol{x})$. Since we are minimizing the loss function $f$, we define the filtering threshold as $\text{UCB}_{f,t,\min} \triangleq \min_{\boldsymbol{x} \in \mathcal{X}} \text{UCB}_{f,t}(\boldsymbol{x})$. Then, the following sublevel-set

$$\hat{\mathcal{X}}^t \triangleq \{\boldsymbol{x} \in \mathcal{X} \mid \text{LCB}_{f,t}(\boldsymbol{x}) \leq \min(\text{UCB}_{f,t,\min}, 0)\} \quad (11)$$

serves as the region(s) of interest (ROI)[2].

## 4.3 EFFICIENT HIGH-DIMENSIONAL OPTIMIZATION THROUGH ROI REDUCTION

Through the optimization, reducing the ROI $\hat{\mathcal{X}}^t$ alleviates the difficulty of learning on the high-dimensional search space. See Figure 1c for an illustration where 10 initialization points have reduced our search space for learning the NE of Example 1. Combined with the following acquisition function, the proposed algorithm ARISE achieves an adaptive trade-off between exploration and exploitation.

$$\alpha_{f,t}(\boldsymbol{x}, \mathcal{S}) = \text{UCB}_{f,t}(\boldsymbol{x}, \mathcal{S}) - \text{LCB}_{f,t}(\boldsymbol{x}, \mathcal{S}) \quad (12)$$

This acquisition differentiates from the well-known variance reduction acquisition function in active learning domain [MacKay, 1992] in twofolds. First, the acquisition function is defined on both confidence intervals of each utility function $u_i$, and the confidence interval tailored to the marginal maximum on $v_i$ as defined in Equation (7) and Equation (8), which are differentiated from the naive definition of the confidence interval on a global Gaussian process. Second, as is shown in the following, we only optimize the acquisition function in a subset of the search space $\hat{\mathcal{X}}^t$ instead of the whole search space $\mathcal{X}$. The reduction of $\hat{\mathcal{X}}^t$ guarantees the efficiency of the optimization by avoiding unnecessary queries in the low utility region.

The ROI identification could be computationally expensive, especially in high-dimensional search space, as it requires point-wise comparison. Thus, its efficiency is highly dependent on the size and distribution of the discretization of the search space. The ROI identification and reduction along the optimization could help mitigate the efficiency problem.

---

[2]In practice, since with high probability $\text{UCB}_{f,t,\min} \geq f^*$, and by assumption the search space consists the NE ($f^* = 0$), it holds that with high probability the ROI threshold is zero.

---

**Algorithm 1** **A**daptive **R**egion of **I**nterest **S**earch for Nash **E**quilibrium (ARISE)

---

1: **Input**: Search space $\mathcal{X}$, initial observation $\mathcal{D}^0$, horizon $T$;
2: **for** $t = 1\ to\ T$ **do**
3:  Fit the Gaussian processes $\mathcal{GP}_{u_i,t}$: $\theta_{u_i,t} \leftarrow \arg\min_{\theta_{u_i}} -\log \mathbb{P}\left[y_i^{1:t-1} \mid \boldsymbol{x}^{1:t-1}, \theta_{u_i}\right]$
4:  Identify ROIs via sublevel-set estimation $\hat{\mathcal{X}}^t \leftarrow \{\boldsymbol{x} \in \mathcal{X} \mid \mathrm{LCB}_{f,t}(\boldsymbol{x}) \leq 0\}$
5:  Optimize the sublevel-set acquisition function: $\boldsymbol{x}^t \leftarrow \arg\max_{\boldsymbol{x} \in \hat{\mathcal{X}}^t} \alpha_{f,t}(\boldsymbol{x}, \hat{\mathcal{X}}^t)$ as in Equation (12)
6:  $\mathcal{D}^{1:t} \leftarrow \mathcal{D}^{1:t-1} \cup \{(\boldsymbol{x}^t, \boldsymbol{y}^t)\}$
7: **end for**
8: **Output**: $\arg\min_{\boldsymbol{x} \in \hat{\mathcal{X}}^T} \mathrm{LCB}_{f,T}(\boldsymbol{x})$

---

In the following section, we offer a theoretical analysis in Lemma 1 showing that the ROI identification in line 4 of Algorithm 1 could be equivalent to

$$\hat{\mathcal{X}}^t = \{\boldsymbol{x} \in \hat{\mathcal{X}}^{t-1} \mid \mathrm{LCB}_{f,t}(\boldsymbol{x}) \leq 0\} \qquad (13)$$

when setting $\hat{\mathcal{X}}^0 = \mathcal{X}$. This means that the ROI identification is actually a hierarchical filtering of the search space and is accelerated by its continuing shrinkage. There is no guarantee of the ROI shrinkage rate, potentially making its performance unstable in High-Dimensional BO (HDBO) tasks. There are several potential solutions. There are chances to incorporate existing orthogonal HDBO techniques, including sparse GP [McIntire et al., 2016, Moss et al., 2023] and dimension reduction for BO [Song et al., 2022, Wang et al., 2016, Letham et al., 2020, Munteanu et al., 2019, Papenmeier et al., 2022]. However, the methods require additional structural assumptions that do not necessarily hold in NE discovery and, therefore, require cautiousness depending on the application.

**Remark.** *The proposed algorithm* ARISE *targets games with discretized strategy spaces for identifying the ROI, similar to previous works by Picheny et al. [2019]. To tackle continuous search space where no smoothness guarantee is known to discretize the space to allow efficient ROI identification. We propose an optional method in the Appendix C to accelerate the candidate pick in the high-dimensional space by formulating the ROI identification and the acquisition function optimization in lines 4 and 5 of Algorithm 1 together as a conventional constrained optimization problem and solve it efficiently with an over-the-shelf tool.*

## 5 THEORETICAL RESULTS

We summarize the required assumptions below, followed by the justification of each assumption.

**Assumption 2.** *The utility functions $u_i$ are sampled from corresponding mutually independent GP. That is, $\forall t \leq T, \boldsymbol{x} \in \mathcal{X}, i \in [n]$, $u_i(\boldsymbol{x})$ is a sample from global $\mathcal{GP}_{u_i,t}$.*

This assumption is commonly found in the literature, as demonstrated by references such as Srinivas et al. [2009],

Gotovos et al. [2013], Zhang et al. [2023]. While devising a well-specified prior for the unknown function could be challenging in practice, there are recent advancements focusing on analyzing BO's behavior under prior misspecification [Bogunovic and Krause, 2021], or proposing solutions for unknown hyperparameters specifying the prior [Berkenkamp et al., 2019, Hvarfner et al., 2024]. Though this is a separate direction orthogonal to our work, we want to highlight the aforementioned challenge and potential for integrating existing solutions.

**Assumption 3.** *Given the horizon $T$, with a proper choice of constant $\beta$, the confidence intervals are well calibrated, meaning a later posterior would agree with the previous posteriors. Concretely, for all $u_i, i \in [n]$. That is, $\forall t_1 \leq t_2 \leq T, \boldsymbol{x} \in \mathcal{X}, i \in [n]$, we have $\mathrm{UCB}_{u_i,t_1}(\boldsymbol{x}) \geq \mathrm{UCB}_{u_i,t_2}(\boldsymbol{x})$ and $\mathrm{LCB}_{u_i,t_1}(\boldsymbol{x}) \leq \mathrm{LCB}_{u_i,t_2}(\boldsymbol{x})$.*

This is a mild assumption given recent work by Koepernik and Pfaff [2021] showing that if the kernel is continuous and the sequence of sampling points lies sufficiently dense, the variance of the posterior $\mathcal{GP}$ converges to zero almost surely monotonically if the function is in metric space, and the posterior mean converges to the unknown function pointwise in $\mathbf{L}^2$ if the unknown function lies in the RKHS of the prior kernel.

If the assumption is violated, the technique of taking the intersection of all historical confidence intervals introduced by Gotovos et al. [2013] could similarly guarantee a monotonically shrinking confidence interval. That is, when $\exists t_1 \leq t_2 \leq T, \boldsymbol{x} \in \mathcal{X}, i \in [n]$, if we have $\mathrm{UCB}_{u_i,t_1}(\boldsymbol{x}) < \mathrm{UCB}_{u_i,t_2}(\boldsymbol{x})$ or $\mathrm{LCB}_{u_i,t_1}(\boldsymbol{x}) > \mathrm{LCB}_{u_i,t_2}(\boldsymbol{x})$, we let $\mathrm{UCB}_{u_i,t_2}(\boldsymbol{x}) = \mathrm{UCB}_{u_i,t_1}(\boldsymbol{x})$ or $\mathrm{LCB}_{u_i,t_2}(\boldsymbol{x}) = \mathrm{LCB}_{u_i,t_1}(\boldsymbol{x})$ to guarantee monotonocity.

A direct result of the assumed monotonously on the confidence interval of $u_i$ is the similar monotonicity on the confidence interval of $v_i$ and $f$, and then the monotonical shrinking of ROI.

**Lemma 1.** *With the Assumption 2 and Assumption 3, $\forall t_1 \leq t_2 \leq T, \boldsymbol{x} \in \mathcal{X}, i \in [n]$, we have $\mathrm{UCB}_{v_i,t_1}(\boldsymbol{x}) \geq \mathrm{UCB}_{v_i,t_2}(\boldsymbol{x})$ and $\mathrm{LCB}_{v_i,t_1}(\boldsymbol{x}) \leq \mathrm{LCB}_{v_i,t_2}(\boldsymbol{x})$. $\forall t_1 \leq t_2 \leq$*

$T, \boldsymbol{x} \in \mathcal{X}$, we have $UCB_{f,t_1}(\boldsymbol{x}) \geq UCB_{f,t_2}(\boldsymbol{x})$ and $LCB_{f,t_1}(\boldsymbol{x}) \leq LCB_{f,t_2}(\boldsymbol{x})$, and therefore $\hat{\mathcal{X}}^t \subseteq \hat{\mathcal{X}}^{t-1}$.

First, we justify the definition of the confidence intervals, and therefore, the ROI identified does not lose the global optimum with a certain probability.

**Lemma 2.** *With the assumptions above, the region(s) of interest $\{\hat{\mathcal{X}}^t\}_{t \in [T]}$ defined in Equation (11) contains the global optimum with high probability. That is, for all $\delta \in (0,1)$, $\forall t \geq 1$, and any finite discretization $\tilde{S}$ of $\mathcal{X}$ containing the optimum $\boldsymbol{x}^* = \arg\min_{\boldsymbol{x} \in \mathcal{X}} f(\boldsymbol{x})$, with $\beta = 2\log(n|\tilde{S}|T/\delta)$, we have $\mathbb{P}\left[\boldsymbol{x}^* \in \hat{\mathcal{X}}^t\right] \geq 1 - \delta$.*

Finally, we bound the simple regret of the proposed Algorithm 1. For clarity, we denote $\tilde{S}_{\hat{\mathcal{X}}^t} = \tilde{S} \cap \hat{\mathcal{X}}^t$, and

$$CI_{f^*,t} = [\min_{\boldsymbol{x} \in \tilde{S}_{\hat{\mathcal{X}}^t}} LCB_{f,t}(\boldsymbol{x}), \min_{\boldsymbol{x} \in \tilde{S}_{\hat{\mathcal{X}}^t}} UCB_{f,t}(\boldsymbol{x})]$$

Let us define the maximum information gain about function $u$ after $T$ rounds:

$$\gamma_{u_i,T} = \max_{A \subset \tilde{S}:|A|=T} \mathbb{I}(y_A; u_i) \text{ and } \widehat{\gamma_T} = \sum_{i \in [n]} \gamma_{u_i,T} \quad (14)$$

Note that previous work by Srinivas et al. [2009] that bounds the maximum information gain $\gamma$ corresponding to popular kernel to be sublinear.

Here, we justify that the proposed acquisition function reduces the width of the confidence interval of the global optimum efficiently.

**Theorem 1.** *The width of the resulting confidence interval of the global optimum $f^* = f(\boldsymbol{x}^*)$ has an upper bound. That is, under the assumptions above, with a constant $\beta = 2\log(n|\tilde{S}|T/\delta)$, and $\boldsymbol{x}^t = \arg\max_{\boldsymbol{x} \in \mathcal{X}} \alpha_{f,t}(\boldsymbol{x}, \mathcal{X})$, after at most $T \geq \frac{\beta \widehat{\gamma}_T \hat{C}_1}{\epsilon^2}$ iterations, we have*

$$\mathbb{P}\left[|CI_{f^*,T}| \leq \epsilon, f^* \in CI_{f^*,T}\right] \geq 1 - \delta$$

*Here $\hat{C}_1 = 8(n+1)^2/\log(1+\sigma^{-2})$.*

The result above shows that when the proposed acquisition function is maximized in the global search space, it achieves efficient learning. However, to reach a balance of exploration and exploitation so that the algorithm identifies the global optimum along with the learning with high probability, we need to restrict the search space to the ROI, which achieves the exploitation by design.

The following results show that, when combining the results above, since the Nash-Equilibrium exists, and the points of ROI are sufficiently close to $\boldsymbol{x}^*$, we have with probability at least $1 - \delta$ that ARISE achieves $\epsilon$-Nash Equilibrium.

**Theorem 2.** *We assume the aforementioned assumptions hold. We apply the same $\beta$ and the acquisition function as illustrated in Algorithm 1. In addition, we assume after $T \geq \frac{\beta \widehat{\gamma}_T \hat{C}_1}{\epsilon^2}$ iterations, when $\forall \boldsymbol{x} \in \tilde{S}_{\hat{\mathcal{X}}^t}$, it holds that $UCB_{u_i,t}(\boldsymbol{x}_{-i}, \tilde{S}_{\hat{\mathcal{X}}^t}) = UCB_{u_i,t}(\boldsymbol{x}_{-i}, \tilde{S})$ and $LCB_{u_i,t}(\boldsymbol{x}_{-i}, \tilde{S}_{\hat{\mathcal{X}}^t}) = LCB_{u_i,t}(\boldsymbol{x}_{-i}, \tilde{S})$, we have*

$$\mathbb{P}\left[f(\boldsymbol{x}^T) \leq \sqrt{\frac{\beta \widehat{\gamma}_T \hat{C}_1}{T}} \leq \epsilon\right] \geq 1 - \delta$$

*Here $\hat{C}_1 = 8(n+1)^2/\log(1+\sigma^{-2})$.*

**Remark.** *The additional assumption made above in Theorem 2 is mild, as it is satisfied when the points in ROI are sufficiently close to the global optimum. This allows that they resemble the Nash Equilibria's property, that is, the partial maximum of the utility functions is identical to $\boldsymbol{x}$ when $f(\boldsymbol{x}) = 0$. More formally, when $\boldsymbol{x} \in \tilde{S}_{\hat{\mathcal{X}}^t}$ converges to $\boldsymbol{x}^*$ where $f(\boldsymbol{x}^*) = 0$, the partial maximum $\arg\max_{\boldsymbol{x} \in \mathcal{X}} v_i(\boldsymbol{x}_{-i})$ also converges to points in ROI.*

Given that $\widehat{\gamma}_T$ and $\beta$ are sublinear to $T$, $\hat{C}_1$ is a constant, the result above shows that the proposed Algorithm 1 achieves $\epsilon$-Nash Equilibria with high probability efficiently.

One direct result of Theorem 2 is that if any point belongs to $\tilde{S}$ that bears a suboptimal gap on the reward except for the global optimum. Then, after sufficient query, the algorithm will identify $\boldsymbol{x}^*$ as the only point in the ROI. In that case, ARISE will only query $\boldsymbol{x}^*$ and achieve zero regret afterward.

**Corollary 1.** *We assume the aforementioned conditions in Theorem 2 hold, and $\forall \boldsymbol{x} \in \tilde{S}$, $\boldsymbol{x} \neq \boldsymbol{x}^*$, it holds that $f(\boldsymbol{x}) > \epsilon$. Then we have*

$$\mathbb{P}\left[f(\boldsymbol{x}^T) = 0\right] \geq 1 - \delta$$

Similarly, if starting from $t'$ before $T$, the ROI only consists of a group of suboptimal candidates that is sufficiently close to $\boldsymbol{x}^*$ and meets the condition assumed in Theorem 2, then the algorithm achieves a sublinear cumulative regret after identifying this near-optimal region, and is therefore no-regret after $t'$.

**Corollary 2.** *We assume the aforementioned conditions in Theorem 2 hold, and $\exists t' < T$ such that $t' \geq \frac{\beta \widehat{\gamma}_t \hat{C}_1}{\epsilon^2}$. Then we have*

$$\mathbb{P}\left[\sum_{t=t'}^{T} f(\boldsymbol{x}^t) \leq \sqrt{T\beta\gamma_T\hat{C}_1}\right] \geq 1 - \delta$$

**Remark.** *The result above shows that Algorithm 1 achieves no regret after identifying the near-optimal region and the cumulative regret is sublinear. Though the analysis assumes*

*a discretization that consists of the Nash Equilibria, the result is also applicable to the continuous version of the problem, as long as the discretization is sufficiently dense and there is an additional smoothness guarantee on the utilities. Then, the density combined with the assumed smoothness could be translated into an approximation error due to the discretization, and the result is still applicable.*

# 6 EXPERIMENTAL RESULTS

We compare the proposed algorithm ARISE with the following baselines. (1) ARISE-GLOBAL removes the ROI identification of ARISE and maximizes the proposed acquisition function globally as discussed in Theorem 1. The comparison serves as an ablation study demonstrating that the introduction of ROI allows ARISE the trade-off of exploration and exploitation rather than pure exploration. (2) We employ PREDICTION and EPSILON GREEDY from Al-Dujaili et al. [2018] with $\epsilon = 0.1$. PREDICTION corresponds to their method using approximated regret as the acquisition function, a pure exploitation subroutine of EPSILON GREEDY. Meanwhile, EPSILON GREEDY achieves the trade-off of exploration and exploitation. The hyper-parameter $\epsilon$ controls the probability of exploration achieved by uncertainty reduction. (3) We compare with SUR (Stepwise Uncertainty Reduction) proposed by Picheny et al. [2019], which is essentially global uncertainty reduction on multiple unknown utility functions. For efficiency, we take advantage of recent advancements in deep kernel learning [Wilson et al., 2016, Zhang et al., 2022] and employ it in both the proposed methods and the baseline.

We examine the performance of our proposed algorithm on the following games.

**Saddle.** This corresponds to the running example we presented in Example 1 and is also discussed by Al-Dujaili et al. [2018], Picheny et al. [2019].

|          | Rock     | Paper    | Scissors  |
|----------|----------|----------|-----------|
| Rock     | (0, 0)   | (-1, 1)  | (1, -1)   |
| Paper    | (1, -1)  | (0, 0)   | (-1, 1)   |
| Scissors | (-1, 1)  | (1, -1)  | (0, 0)    |

Table 1: Payoffs of the rock-paper-scissors game. Each utility element $(i, j)$ means the row agent receives $i$ utility and the column agent receives $j$.

**Rock-Paper-Scissors (RPS).** In this game, two agents' strategies are denoted by $x_1, x_2 \in \Delta^2 = \{x \in \mathbb{R}^3 : x^r + x^p + x^s = 1\}$, and the utilities are defined as

$$u_1(x_1, x_2) = (x_1^p - x_1^s)x_2^r + (x_1^s - x_1^r)x_2^p + (x_1^r - x_1^p)x_2^s,$$
$$u_2(x_1, x_2) = (x_2^p - x_2^s)x_1^r + (x_2^s - x_2^r)x_1^p + (x_2^r - x_2^p)x_1^s. \quad (15)$$

The NE is attained at $x_1 = x_2 = (1/3, 1/3, 1/3)$.

**Hotelling's Game.** We explore another classical structured game with real-world applications [Brenner, 2005]. Imagine a market where two firms must choose their locations on a 2-$d$ grid to attract customers. Each firm wants to attract customers, and the utility depends on the number of customers they draw. The firms have to balance being close to customers while avoiding excessive competition. Let us consider the total area as a unit square, and each firm's action is to choose location $x = (x^N, x^W) \in [0, 1]^2$. We assume the customer population is uniformly distributed over the total area, and two firms post the same price for the products. Therefore, a customer prefers a firm that is close by. Given the two firms' actions $(x_1^N, x_1^W)$ and $(x_2^N, x_2^W)$, their utility can be computed by the area of agents whose distance is closer to themselves than the competitor. For example, let $S_1 = \{(x^N, x^W) | (x^N - x_1^N)^2 + (x^W - x_1^W)^2 \leq (x^N - x_2^N)^2 + (x^W - x_2^W)^2\}$ and firm 1 utility is $S_1$'s area.

**Marketing Budget Allocation Game.** Finally, we present a real-world marketing problem, where advertisers seek to maximize the number of customers by allocating given budgets to each media channel effectively [Maehara et al., 2015]. Let $G = (S \cup Z, E)$ be a bipartite graph, where the left vertices $S$ denote media channels, the right vertices $Z$ denote customers, and the edges $E \subseteq S \times Z$ denote the relations between channels and customers. Each edge $(s, z) \in E$ has an activation probability $p(s, z) \in [0, 1]$ such that customer $z \in Z$ is activated via channel $s \in S$ with probability $p(s, z)$.

There are $n$ advertisers, where each advertiser's strategy is $x_i \in \mathbb{N}_{\geq 0}^{|S|}$ denotes a vector of allocated units for $|S|$ channels. The strategy space for each advertiser is

$$X_i = \{x_i \in \mathbb{N}_{\geq 0}^{|S|} : x_i(s) \leq c(s) \, \forall s; \langle w, x_i \rangle \leq B\},$$

where $c(s)$ denotes the capacity of every channel and $w \in \mathbb{R}_+^{|S|}$ denotes the cost of every unit for all channels. Let $\Sigma_n$ denote the set of all permutations of $[n]$. Finally, the utility of every advertiser $i \in [n]$ is denoted as

$$u_i(\boldsymbol{x}) = \frac{1}{n!} \sum_{z \in Z} \sum_{\sigma \in \Sigma_n} P_i(x_i, z) \prod_{j \prec_\sigma i} \left(1 - P_j(x_j, z)\right) \quad (16)$$

where $P_i(x_i, z) = 1 - \prod_{s \in S}(1 - p(s, z))^{x_i(s)}$ denotes the probability of customer $z$ being activated by advertiser $i$ under the units allocation plan $x_i$. In the experiment, we set $n = 2$, $|S| = 4$ and $|Z| = 12$.

**Discussion.** As is shown in Figure 2, ARISE consistently matches or outperforms the baselines. The comparison with ARISE-GLOBAL shows that the introduced ROI identification significantly contributes to the general performance. Though implemented differently, ARISE-GLOBAL and SUR both lack exploitation. Their simple regrets platform at high values in Figure 2 (b), (c), and (d) indicate

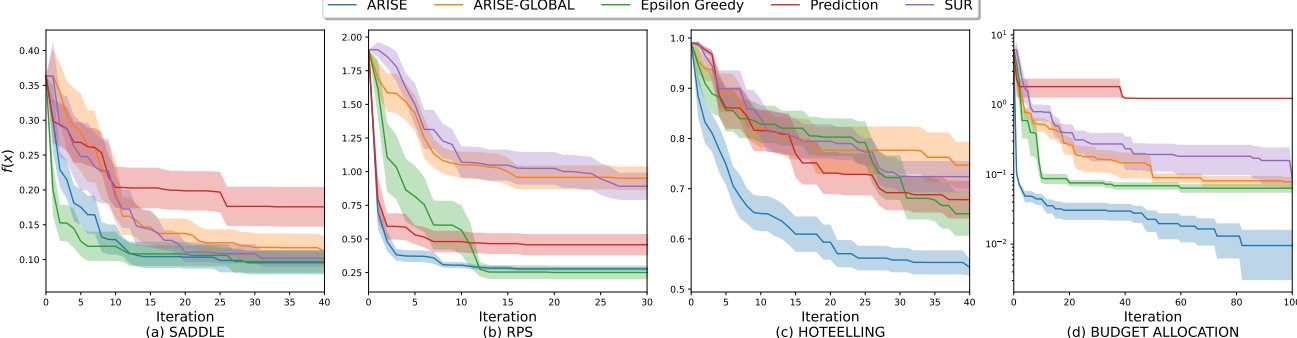

Figure 2: Experimental results. In each plot, the $x$-axis denotes the number of function evaluations. The curves show the $f(\boldsymbol{x}^t)$ values averaged over at least ten independent trials. The shaded area denotes the standard error. The observation perturbation is sampled from $\mathcal{N}(0, 0.01)$, while the simple regrets shown in the figures do not count the noise. We also include additional results on multi-player settings in Appendix E.

the intrinsic complexity of the corresponding problems. EPSILON GREEDY outperforms PREDICTION in Figure 2(a), (c), and (d), showing the importance of the trade-off of exploration and exploitation in the learning process. ARISE outperform EPSILON GREEDY in Figure 2(c) and (d) showing that in complex setting, ARISE achieves a principled and more efficient trade-off.

## 7 CONCLUSIONS

We study the problem of learning Nash equilibrium of black-box games with a Bayesian approach using Gaussian processes as surrogates for the unknown utilities. We characterize the equilibrium computation problem as optimizing an unknown objective function. As a result, finding the Nash equilibrium of the game is equivalent to minimizing the unknown objective function. We also proposed a no-regret learning approach to minimize the unknown objective function with principled ROI identification and acquisition maximization. Our study shows the proposed algorithm improves upon existing methods both with novel theoretical results and strong empirical performance across various tasks.

Our results open the possibilities for many other interesting questions. For example, our work and prior research primarily address learning NE in normal-form games, where agents act simultaneously. Another intriguing domain is Stackelberg games, where agents move sequentially (cf. Appendix A). Hence, exploring Stackelberg equilibrium computation presents another interesting problem to investigate. Furthermore, we assume the GPs of distinct agents are independent. Investigating the correlation between agents' utility functions and constructing multivariate GPs presents an intriguing avenue for future exploration as well.

## Acknowledgements

Minbiao Han is supported in part by the Army Research Office Award W911NF-23-1-0030 and the Office of Naval Research Award N00014-23-1-2802. Yuxin Chen and Fengxue Zhang acknowledge support through grants from the National Science Foundation under Grant No. NSF CMMI-2037026 and NSF IIS-2313130.

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

# Supplementary Material

**Minbiao Han**[*1]          **Fengxue Zhang** [*1]          **Yuxin Chen**[1]

[1]Department of Computer Science, University of Chicago, Chicago, Illinois, USA

## A  ADDITIONAL RELATED WORK

### A.1  LEARNING STACKELBERG GAMES

More broadly, our research is also related to learning the equilibrium in the game theory. Besides the Nash equilibrium studied in this paper, another well-studied game is the Stackelberg game [Von Stengel and Zamir, 2004, Gan et al., 2023]. Specifically, Stackelberg games model a two-step sequential decision-making process between two agents, a leader and a follower. This canonical model for strategic leader-follower interactions has been adopted for many applications in the real world, such as contract design, optimal pricing, security resource allocation, and optimal traffic routing [Bolton and Dewatripont, 2004, Dawkins et al., 2021, Paruchuri et al., 2008, Roth et al., 2016, Roughgarden, 2001, Yang et al., 2014]. Learning the Stackelberg equilibrium has also been extensively studied in the literature [Letchford et al., 2009, Peng et al., 2019, Dawkins et al., 2022, Han et al., 2024], it would be interesting to study the learning of Stackelberg game equilibria via Gaussian Processes.

### A.2  BO WITH MULTIPLE STRUCTURED UTILITY FUNCTIONS

Within the scope of Bayesian optimization tasks, it is common to tackle multiple unknowns, as in the learning of equilibria, where the algorithm needs to deal with multiple unknown utility functions. The most related literature in the realm of BO would be optimizing the function network, where the objective function to be optimized could be decomposed into multiple unknown nodes in a known directed acyclic graph [Astudillo and Frazier, 2021, Buathong et al., 2023]. Similarly, Sussex et al. [2022] proposes to optimize the intervention on the casual graph with the extension of UCB, a canonical acquisition in BO, and offers a corresponding theoretical guarantee on the convergence. These works assume that each node on the DAG graph representing the unknown function could be captured by a separate GP and assume independence between different nodes. However, when transferring our objective into a DAG, we are dealing with highly related nodes as will be illustrated in the following section. The reason is that part of the components of the ultimate objective is the partial maximization of the other. Also, unlike in the graph-based BO works, we would not observe the partial maximization and, therefore, could not update the GPs for all the nodes with corresponding observations. The gap in the assumption and process of evaluation hinders the direct application of the graph-based BO methods.

---

[*]Equal Contribution, the author names are in alphabetical order.

# B PROOFS

## B.1 PROOF OF LEMMA 2

*Proof.* Similar to lemma 5.1 of Srinivas et al. [2009], given a constant $\beta = 2\log(n|\tilde{S}|T/\delta)$, with probability at least $1 - \delta$, $\forall \boldsymbol{x} \in \tilde{S}, \forall t \geq 1, \forall g \in \{u_i\}_{i \in [n]} \cup \{v_i\}_{i \in [n]}$,

$$|g(\boldsymbol{x}) - \mu_{g,t-1}(\boldsymbol{x})| \leq \beta^{1/2}\sigma_{g,t-1}(\boldsymbol{x})$$

Note that we also take the union bound on $g \in \{u_i\}_{i \in [n]} \cup \{v_i\}_{i \in [n]}$.

Then, we have $\forall t \leq T, \boldsymbol{x} \in \tilde{S}$

$$\mathbb{P}\left[\text{UCB}_{v_i,t}(\boldsymbol{x}) = \max_{x_i'}\text{UCB}_{u_i,t}(x_i', \boldsymbol{x}_{-i}) \geq \max_{x_i'} u_i(x_i', \boldsymbol{x}_{-i}) = v_i(\boldsymbol{x}_{-i})\right] \geq 1 - \delta$$

and at the same time

$$\mathbb{P}\left[\text{LCB}_{v_i,t}(\boldsymbol{x}) = \max_{x_i'}\text{LCB}_{u_i,t}(x_i', \boldsymbol{x}_{-i}) \leq \max_{x_i'} u_i(x_i', \boldsymbol{x}_{-i}) = v_i(\boldsymbol{x}_{-i})\right] \geq 1 - \delta$$

This justifies the definition of Equation (7) and Equation (8).

As a result, we also have $\forall t \leq T, \boldsymbol{x} \in \tilde{S}$

$$\mathbb{P}\left[\text{UCB}_{f,t}(\boldsymbol{x}) \geq f(\boldsymbol{x}) \geq f(\boldsymbol{x}^*) \geq \text{LCB}_{f,t}(\boldsymbol{x}^*)\right] \geq 1 - \delta$$

By the definition of the threshold $\text{UCB}_{f,t,\min}$ we have $\forall t \leq T$,

$$\mathbb{P}\left[\text{UCB}_{f,t,\min} > \text{LCB}_{f,t}(\boldsymbol{x}^*)\right] \geq 1 - \delta$$

By the definition of the $f(\boldsymbol{x})$, we have $\forall \boldsymbol{x}, f(\boldsymbol{x}) \geq 0$.

Hence we have $\forall t \leq T, \forall i \in [n]$

$$\mathbb{P}\left[\boldsymbol{x}^* \in \hat{\mathcal{X}}^t\right] \geq 1 - \delta$$

$\square$

## B.2 PROOF OF THEOREM 1

*Proof.* The following proof shows that the width of the interval at $t$ is bounded. For briefness, we denote $\alpha_t \triangleq \max_{\boldsymbol{x} \in \mathcal{X}} \alpha_{f,t}(\boldsymbol{x}, \mathcal{X})$

With probability at least $1 - \delta, \forall T \geq t \geq 1$, we first have

$$f(\boldsymbol{x}^*) \in [\text{LCB}_{f,t,\min}, \text{UCB}_{f,t,\min}]$$

and then

$$\text{UCB}_{f,t,\min} - \text{LCB}_{f,t,\min} \leq \alpha_t$$

By lemma 5.1, 5.2 and 5.4 of Srinivas et al. [2009], with $\beta = 2\log(n|\tilde{S}|T/\delta)$, $\forall g \in \{u_i\}_{i \in [n]}$, we have

$\sum_{t=1}^{T}(2\beta^{1/2}\sigma_{g,t-1}, (\boldsymbol{x}^t))^2 \leq C_1\beta\gamma_{g,T}$. Then we have the following hold with probability at least $1 - \delta$:

$$\sum_{t=1}^{T}\alpha_t^2 \leq \sum_{t=1}^{T}(\mathrm{UCB}_{f,t}(\boldsymbol{x}^t) - \mathrm{LCB}_{f,t}(\boldsymbol{x}^t))^2$$

$$\leq \sum_{t=1}^{T}((n+1)\sum_{g\in\{u_i\}_{i\in[n]}}2\beta^{1/2}\sigma_{g,t-1}(\boldsymbol{x}^t))^2$$

$$= (n+1)^2\sum_{t=1}^{T}\sum_{g\in\{u_i\}_{i\in[n]}}(2\beta^{1/2}\sigma_{g,t-1}(\boldsymbol{x}^t))^2$$

$$\leq (n+1)^2\sum_{g\in\{u_i\}_{i\in[n]}}C_1\beta\gamma_{g,T}$$

$$= (n+1)^2C_1\beta\widehat{\gamma}_T$$

Where $C_1 = 8/\log(1 + \sigma^{-2})$. The second line holds for two reasons. First, we have $\forall g \in \{u_i\}_{i\in[n]}$, $\mathrm{UCB}_{g,t}(\boldsymbol{x}^t) - \mathrm{LCB}_{g,t}(\boldsymbol{x}^t) \leq 2\beta^{1/2}\sigma_{g,t-1}(\boldsymbol{x}^t)$. Also, we have $\forall g \in \{v_i\}_{i\in[n]}$, $\mathrm{UCB}_{g,t}(\boldsymbol{x}^t) - \mathrm{LCB}_{g,t}(\boldsymbol{x}^t) \leq \sum_{i\in[n]}\mathrm{UCB}_{u_i,t}(\boldsymbol{x}^t) - \mathrm{LCB}_{u_i,t}(\boldsymbol{x}^t)$ since $\boldsymbol{x}^t$ maximize $\alpha_{f,t}$. The last line holds due to the definition in Equation (14). By Cauchy-Schwarz, we have with probability at least $1 - \delta$:

$$\frac{1}{T}(\sum_{t=1}^{T}\alpha_t)^2 \leq (n+1)^2C_1\beta\widehat{\gamma}_T$$

By the monotonocity assumed in *Assumption 3*, $\forall g \in [n]$, $\forall 1 \leq t_1 < t_2 \leq T$, we have $\alpha_{t_2} \leq \alpha_{t_1}$. Therefore with probability at least $1 - \delta$:

$$|CI_{f^*,T}| \leq = \mathrm{UCB}_{f,T,\min} - \mathrm{LCB}_{f,T,\min}$$

$$\leq \alpha_T$$

$$\leq \sqrt{\frac{(n+1)^2\beta C_1\widehat{\gamma}_T}{T}}$$

For briefness, we denote $\hat{C}_1 = 8(n+1)^2/\log(1+\sigma^{-2})$, then as long as $T \geq \frac{\beta\widehat{\gamma}_T\hat{C}_1}{\epsilon^2}$, we have with probability at least $1-\delta$

$$|CI_{f^*,T}| \leq \epsilon$$

$\square$

## B.3 PROOF OF THEOREM 2

The following results bound the simple regret of the proposed Algorithm 1 with additional mild assumptions.

Different from the proof of Theorem 1, we are optimizing the acquisition on the ROI rather than the global search space. The key insight that

$$\sum_{t=1}^{T}\alpha_t^2 \leq \sum_{t=1}^{T}(\mathrm{UCB}_{f,t}(\boldsymbol{x}^t) - \mathrm{LCB}_{f,t}(\boldsymbol{x}^t))^2$$

$$\leq \sum_{t=1}^{T}((n+1)\sum_{g\in\{u_i\}_{i\in[n]}}2\beta^{1/2}\sigma_{g,t-1}(\boldsymbol{x}^t))^2$$

no longer holds. Instead, we can only bound for $\hat{\alpha}_t \triangleq \max_{\boldsymbol{x} \in \tilde{S}_{\hat{\mathcal{X}}^t}} \alpha_{f,t}(\boldsymbol{x}, \tilde{S}_{\hat{\mathcal{X}}^t})$ similarly.

$$\sum_{t=1}^{T} \hat{\alpha}_t^2 = \sum_{t=1}^{T} (\text{UCB}_{f,t}(\boldsymbol{x}^t, \tilde{S}_{\hat{\mathcal{X}}^t}) - \text{LCB}_{f,t}(\boldsymbol{x}^t, \tilde{S}_{\hat{\mathcal{X}}^t}))^2$$

$$\leq \sum_{t=1}^{T} ((n+1) \sum_{g \in \{u_i\}_{i \in [n]}} 2\beta^{1/2} \sigma_{g,t-1}(\boldsymbol{x}^t))^2$$

Similarly, by Cauchy-Schwarz, we have

$$\sum_{g \in \{u_i\}_{i \in [n]}} \text{UCB}_{g,t}(\boldsymbol{x}^t) - \text{LCB}_{g,t}(\boldsymbol{x}^t) \leq \sqrt{\beta C_1 \widehat{\gamma_T} T}$$

Where $C_1 = 8/\log(1 + \sigma^{-2})$. And with the assumed monotonicity, we have with probability at least $1 - \delta$:

$$\hat{\alpha}_T \triangleq \max_{\boldsymbol{x} \in \tilde{S}_{\hat{\mathcal{X}}^t}} \text{UCB}_{f,t}(\boldsymbol{x}, \tilde{S}_{\hat{\mathcal{X}}^t}) - \text{LCB}_{f,t}(\boldsymbol{x}, \tilde{S}_{\hat{\mathcal{X}}^t})$$

$$\leq \sqrt{\frac{(n+1)^2 \beta C_1 \widehat{\gamma_T}}{T}}$$

Since we are assuming that after $T \geq \frac{\beta \widehat{\gamma_T} \hat{C}_1}{\epsilon^2}$ iterations, $\forall \boldsymbol{x} \in \tilde{S}_{\hat{\mathcal{X}}^t}$, it holds that $\text{UCB}_{u_i,t}(\boldsymbol{x}_{-i}, \tilde{S}_{\hat{\mathcal{X}}^t}) = \text{UCB}_{u_i,t}(\boldsymbol{x}_{-i}, \tilde{S})$ and $\text{LCB}_{u_i,t}(\boldsymbol{x}_{-i}, \tilde{S}_{\hat{\mathcal{X}}^t}) = \text{LCB}_{u_i,t}(\boldsymbol{x}_{-i}, \tilde{S})$, we have $\alpha_T = \hat{\alpha}_T \leq \sqrt{\frac{(n+1)^2 \beta \widehat{\gamma_T} C_1}{T}} = \sqrt{\frac{\beta \widehat{\gamma_T} \hat{C}_1}{T}} \leq \epsilon$.

In summary, we have with probability at least $1 - \delta$:

$$f(\boldsymbol{x}^T) \leq \text{UCB}_{f,T,\min} \leq \sqrt{\frac{\beta \widehat{\gamma_T} \hat{C}_1}{T}} \leq \epsilon$$

## C   EFFICIENT CONSTRAINED OPTIMIZATION

We propose to accelerate the candidate pick in the high-dimensional space by formulating the ROI identification and the acquisition function optimization in lines 4 and 5 of Algorithm 1 together as a conventional constrained optimization problem and solve it efficiently with an over-the-shelf tool.

We first solve the $\text{UCB}_{f,t,\min}$,

$$\text{UCB}_{f,t,\min} = \min_{\boldsymbol{x} \in \mathcal{X}} \text{UCB}_{f,t}(\boldsymbol{x}) \quad \text{s.t.} \quad \text{LCB}_{f,t-1}(\boldsymbol{x}) \leq \text{UCB}_{f,t-1,\min}$$

then identify the candidate $\boldsymbol{x}^t$ to be evaluated:

$$\boldsymbol{x}^t = \arg \max_{\boldsymbol{x} \in \mathcal{X}} \alpha_{f,t}(\boldsymbol{x}, \mathcal{X}) \quad \text{s.t.} \quad \text{LCB}_{f,t}(\boldsymbol{x}) \leq 0$$

Since the above calculation of $\alpha_{f,t}(\boldsymbol{x}, \hat{\mathcal{X}}^t)$ requires a marginal maximum of $\text{UCB}_{v_i,t}$ and $\text{LCB}_{v_i,t}$ for each agent $i \in [n]$, making the optimization a nested optimization problem, we propose the following approximation inspired by the reparametrization trick by Sussex et al. [2022]:

$$\text{UCB}_{v_i,t}(\boldsymbol{x}_{-i}, \mathcal{X}) = \zeta_{i,t,\text{UCB}}(\boldsymbol{x}) \max_{\boldsymbol{x} \in \mathcal{X}} \text{UCB}_{u_i,t}(\boldsymbol{x})$$

$$\text{LCB}_{v_i,t}(\boldsymbol{x}_{-i}, \mathcal{X}) = \text{UCB}_{v_i,t}(\boldsymbol{x}_{-i}, \mathcal{X}) - \zeta_{i,t,\text{LCB}}(\boldsymbol{x}) 2\beta^{1/2} \max_{\boldsymbol{x} \in \mathcal{X}} \sigma_{u_i,t-1}(\boldsymbol{x})$$

where $\zeta_{i,t,\text{UCB}}(\boldsymbol{x}) \in [0,1]$ and $\zeta_{i,t,\text{LCB}}(\boldsymbol{x}) \in [0,1]$ are learned with regression models(e.g. a neural network) that allows gradient-based optimization to optimize with respect to $\boldsymbol{x}$. Here, $\max_{\boldsymbol{x} \in \mathcal{X}} \sigma_{u_i,t-1}(\boldsymbol{x})$ and $\max_{\boldsymbol{x} \in \mathcal{X}} \text{UCB}_{u_i,t}$ are easy to obtain by applying over-the-shelf optimizer on the posterior. The regression models could be trained on related scenarios where the utility functions are known or cheap to evaluate so that the Gaussian process could be updated arbitrarily without incurring significant costs for training models for $\zeta_{i,t,\text{UCB}}$ and $\zeta_{i,t,\text{LCB}}$.

# D CHOICE OF $\beta$

We follow the convention from Srinivas et al. [2009] that applies practical $\beta$ values different from the theoretical results to achieve better empirical performance. We choose $\beta = 1$ for Hotelling and $\beta = 2$ otherwise. We showcase the sensitivity of the choice of $\beta$ for ARISE in Figure 3. Note that though we choose $\beta$ different from theoretical results in Theorem 1 where $\delta = 0.05$, unlike typical hyper-parameters, each choice of value corresponds to a different confidence level of the error bound.

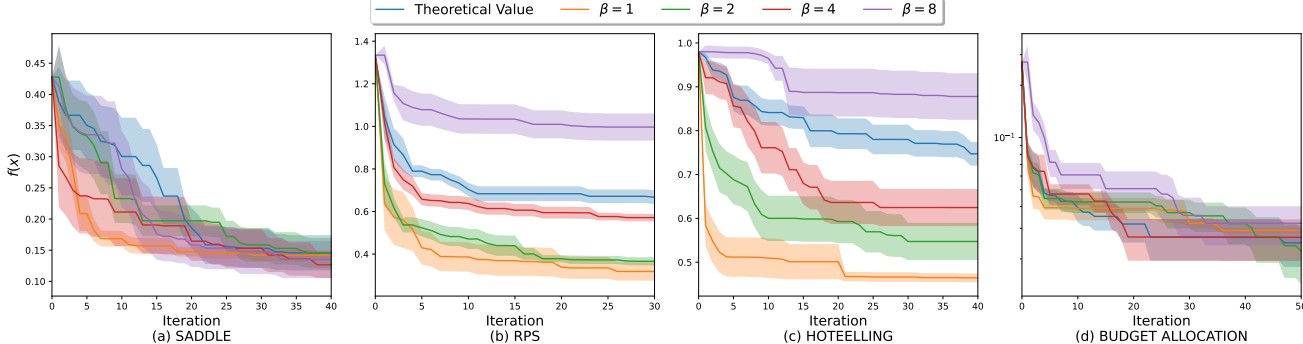

Figure 3: Experimental results on choices of $\beta$. The theoretical value is defined as in Theorem 1. In each plot, the $x$-axis denotes the number of function evaluations. The curves show the $f(\boldsymbol{x}^t)$ values averaged over at least ten independent trials. The shaded area denotes the standard error. The observation perturbation is sampled from $\mathcal{N}(0, 0.01)$, while the simple regrets shown in the figures do not count the noise.

# E ADDITIONAL RESULTS ON 3-PLAYER GAMES

In the following, we incorporate additional experimental results for the Hotelling and Budget Allocation games, specifically examining scenarios with three players.

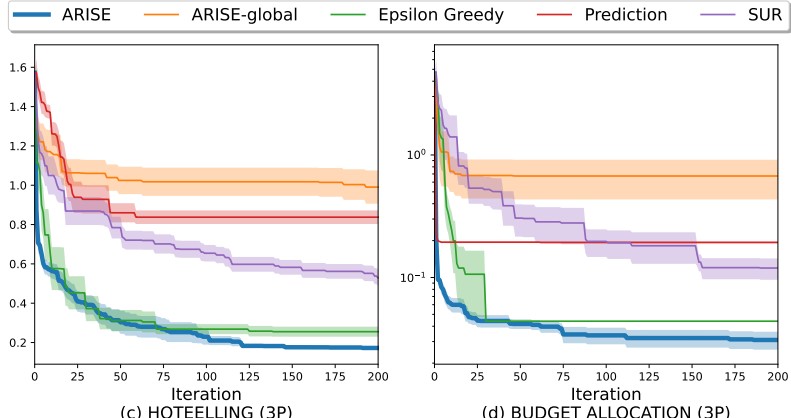

Figure 4: Experimental results on Hotelling and Budget Allocation games when there are 3 players involved, where the $x$-axis denotes the number of function evaluations. The curves show the $f(\boldsymbol{x}^t)$ values averaged over at least ten independent trials, and the shaded area denotes the standard error. The observation perturbation is sampled from $\mathcal{N}(0, 0.01)$, while the simple regrets shown in the figures do not count the noise. The theoretical value is defined as in Theorem 1.

Consistent with our previous results, Figure 4 shows that ARISE outperforms or at least matches the performance of the best baseline method.