# OpenReview forum: "No-Regret Learning of Nash Equilibrium for Black-Box Games via Gaussian Processes"
_auai.org/UAI/2024/Conference — UAI 2024 poster_

### Official Review · Reviewer_y89k · 2024-02-26

**Q2-1 Originality-Novelty:** 2
**Q2-2 Correctness-Technical Quality:** 3
**Q2-5 Clarity Of Writing:** 3

**Q10 Ethical Concerns:**

The submission does not raise potential ethical concerns.

**Q1 Summary And Contributions:**

This paper considers the problem of computing the NE of a game in a black-box manner, i.e., the learner can only have access to the values of utility functions, but not its function information. This is done by this work by combining the problem formulation of [Al-Dujaili et al., 2018] and the optimization method of [Zhang et al., 2023]. Specifically, this work first transforms the problem of solving NE into the problem of minimizing the maximum gain. When the maximum gain equals zero, it means that the agents have come to an NE. Second, optimizing on the whole space is inefficient. To this end, the authors follow the idea of [Zhang et al., 2023] to reduce the ROI space by the relationship between UCB and LCB. The authors have achieved an $O(1/\sqrt{T})$-approximate NE after $T$ iterations. Finally, empirical studies validate the effectiveness of the proposed method.

**Q2-3 Extent To Which Claims Are Supported By Evidence:**

3: Good: the main claims are supported by convincing evidence (in the form of adequate experimental evaluation, proofs, (pseudo-)code, references, assumptions).

**Q2-4 Reproducibility:**

3: Good: key resources (e.g. proofs, code, data) are available and key details (e.g. proofs, experimental setup) are sufficiently well-described for competent researchers to confidently reproduce the main results.

**Q3 Main Strengths:**

This paper solves NEs from a novel perspective by using Gaussian processes and Bayesian optimization techniques. The presentations are clear, intuitive, and thus easy to follow for new readers. This paper obtains the first theoretical guarantee for solving NEs using GP. The empirical studies are sufficient.

**Q4 Main Weakness:**

I do not see major weaknesses in this paper. For remaining questions, please refer to the next part.

**Q5 Detailed Comments To The Authors:**

The presentations are clear. And thus I do not have comments on this point.

Questions:
1. This paper assumes that the utility function follows a Gaussian distribution. Is this assumption much too strong?
2. Is the optimization problem in Line 5 of Algorithm 1 easy to solve? Maybe I have missed the corresponding statements in the paper. However, from my point of view, the efficiency of solving this problem should also be considered.

**Q9 Complying With Reviewing Instructions:**

Yes

---

> ### Author Rebuttal · Authors · 2024-04-03
>
> We thank the reviewer for the insightful comments and for appreciating our results. Below, we clarify the reviewer's primary concerns and will incorporate the discussion into the next version of our work as well.
>
> > This paper assumes that the utility function follows a Gaussian distribution. Is this assumption much too strong?
>
> We follow the **typical assumptions** in the Bayesian Optimization literature (references 1, 2, 3). While we agree with the reviewer that devising a well-specified prior for the unknown could be challenging under specific scenarios, we believe recent advancement has studied BO’s behavior under prior misspecification or proposed solutions to unknown hyperparameters (references 4,5,6). Though this is a separate direction orthogonal to our work and is probably outside the scope of our work, we’d like to highlight the aforementioned challenge and potential for integrating existing solutions in the discussion section.
>
> ***Reference***
>
> 1. Shahriari, Bobak, et al. "Taking the human out of the loop: A review of Bayesian optimization." Proceedings of the IEEE 104.1 (2015): 148-175.
>
> 2. Srinivas, Niranjan, et al. "Gaussian process optimization in the bandit setting: No regret and experimental design." arXiv preprint arXiv:0912.3995 (2009).
>
> 3. Chowdhury, Sayak Ray, and Aditya Gopalan. "On kernelized multi-armed bandits." International Conference on Machine Learning. PMLR, 2017.
>
> 4. Bogunovic, Ilija, and Andreas Krause. "Misspecified gaussian process bandit optimization." Advances in Neural Information Processing Systems 34 (2021): 3004-3015.
>
> 5. Hvarfner, Carl, et al. "Self-correcting bayesian optimization through bayesian active learning." Advances in Neural Information Processing Systems 36 (2024).
>
> 6. Berkenkamp, Felix, Angela P. Schoellig, and Andreas Krause. "No-regret bayesian optimization with unknown hyperparameters." Journal of Machine Learning Research 20.50 (2019): 1-24.
>
>
> > Is the optimization problem in Line 5 of Algorithm 1 easy to solve? Maybe I have missed the corresponding statements in the paper. However, from my point of view, the efficiency of solving this problem should also be considered.
>
> We agree with the reviewer that line 5 of algorithm 1 requires non-trivial optimization and comparison of UCB v.s. LCB, which could potentially be computationally intensive in nature. Yet we want to highlight that (1) such comparison is ***common*** in Active Learning/Bayesian Optimization literature (references 1, 2 ); (2) addressing the computation cost typically requires ***heuristics or additional assumptions*** on the underlying functions, which we believe is beyond the scope of our work (reference 3, 4).
>
> ***Reference***
>
> 1. Bogunovic, Ilija, et al. "Truncated variance reduction: A unified approach to bayesian optimization and level-set estimation." Advances in neural information processing systems 29 (2016).
>
> 2. Zhang, Fengxue, et al. "Learning regions of interest for Bayesian optimization with adaptive level-set estimation." International Conference on Machine Learning. PMLR, 2023.
>
> 3. Eriksson, David, et al. "Scalable global optimization via local Bayesian optimization." Advances in neural information processing systems 32 (2019).
>
> 4. Binois, Mickael, and Nathan Wycoff. "A survey on high-dimensional Gaussian process modeling with application to Bayesian optimization." ACM Transactions on Evolutionary Learning and Optimization 2.2 (2022): 1-26.

---

### Official Review · Reviewer_AgNo · 2024-03-14

**Q2-1 Originality-Novelty:** 3
**Q2-2 Correctness-Technical Quality:** 3
**Q2-5 Clarity Of Writing:** 4

**Q1 Summary And Contributions:**

This article proposes an approach based on Gaussian processes for learning Nash equilibria in games, using only queries of a black-box utility function (action spaces are generally continuous, if not the problem would be trivial).
This provides a "centralized training / decentralized execution" for the learned policies.
The article provides no-regret theoretical guarantees for the learning method, as well as an experimental evaluation.

**Q2-3 Extent To Which Claims Are Supported By Evidence:**

4: Excellent: all claims are supported by very convincing evidence (in the form of comprehensive experimental evaluation, rigorous mathematical proofs, detailed (pseudo-)code, precise references, well-motivated and realistic assumptions) and the authors deliver what they promise.

**Q2-4 Reproducibility:**

4: Excellent: key resources (e.g. proofs, code, data) are available and key details (e.g. proof sketches, experimental setup) are comprehensively described for competent researchers to confidently and easily reproduce the main results.

**Q3 Main Strengths:**

1) The paper is wel-written.
2) The results are theoretically grounded
3) The codes are provided, allowing reproducibility

**Q4 Main Weakness:**

The only weak point I identified is that most of the considered games are two-player zero-sum games, which exhibit particular properties (in the finite-state complete information case, they can be solved in polynomial time). The last game studied is also 2-player (in experiments), but I cannot figure out whether it is "constant-sum" (thus potentially as easy as zero-sum) or not.

**Q5 Detailed Comments To The Authors:**

I found this paper interesting and quite easy to read, given the difficulty of the theoretical results (I must admit I did not check all the proofs).
The results are interesting and potentially useful.
One aspect bothered me, though. The potential scope of the method is unclear to me. The method is tested on 2-player (potentially onlyzero-sum), continuous action space games. The efficiency of the approach on more general games is unclear to me.
1) The approach is not evaluated on n-player continuous action space games.
2) Finite action spaces games can be seen, of course, as continuous action space games on the polytope of probability distributions over strategies. But, on the other hand, if one is to measure "exploitability" of strategies (i.e. loss), it is only necessary to evaluate deviations of strategies to "pure" strategies (due to linearity of expected utility of mixed strategies). Thus, I do not know if the approach is meaningful for finite action spaces and competitive with other learning approaches.

**Q9 Complying With Reviewing Instructions:**

Yes

---

> ### Author Rebuttal · Authors · 2024-04-03
>
> We thank the reviewer for the encouraging feedback and for appreciating our results. Below, we further clarify two points in the reviewer’s comments:
>
> > The approach is not evaluated on n-player continuous action space games.
>
> We appreciate the reviewer for highlighting a point that may have been confusing. We would like to clarify that our paper considers **$n$-player normal-form games** rather than being limited to 2-player zero-sum games. Our theoretical analysis does not rely on the assumptions of either 2-player or zero-sum settings. Furthermore, the experiments we conducted include the Hotelling game, which is a constant-sum game, and the budget allocation game, which models a realistic problem with a more complex utility structure than a constant sum [Maehara et al., 2015].
>
> Regarding the action space, we believe there may have been a misunderstanding. We would like to highlight that the first three games in our experiments all feature **continuous action spaces**. In the Saddle game, an agent's action space is any $x \in [0, 1]$. For the Rock-Paper-Scissors (RPS) game, an agent's action space is represented by a probabilistic vector $x \in \mathbb{R}^3$. Lastly, in the Hotelling game, an agent's action space is the location $x \in \mathbb{R}^2$.
>
> Finally, we appreciate the reviewer's feedback on our careless choice of experimental settings for the 2-player case. Though the experiments are designed to corroborate the theoretical results on simple tasks, we acknowledge that **a broader selection of the tasks could further enhance our results**. We are currently in the process of running additional experiments on the 3-player setting and will strive to include some of these results by the end of the rebuttal period. In any case, we will ensure that more diverse settings are added in the revised version of our paper.
>
>
> > Thus, I do not know if the approach is meaningful for finite action spaces and competitive with other learning approaches.
>
>
> We want to draw attention to the fact that in our final experiment, the Marketing Budget Allocation Game, the action space is finite. It is important to note that our benchmark algorithm, Stepwise Uncertainty Reduction (SUR), proposed by Picheny et al. [2019], was designed for games with either an originally discrete action space or games where a representative discretization is available. This means that our algorithm showcases superior performance on discrete games when compared to learning approaches specifically designed for discretized games.
>
> **References**:
> - Maehara, Takanori, Akihiro Yabe, and Ken-ichi Kawarabayashi. "Budget allocation problem with multiple advertisers: A game theoretic view." International Conference on Machine Learning. PMLR, 2015.
> - Picheny, Victor, Mickael Binois, and Abderrahmane Habbal. "A Bayesian optimization approach to find Nash equilibria." Journal of Global Optimization 73 (2019): 171-192.

---

### Official Review · Reviewer_SHCD · 2024-03-20

**Q2-1 Originality-Novelty:** 2
**Q2-2 Correctness-Technical Quality:** 3
**Q2-5 Clarity Of Writing:** 2

**Q1 Summary And Contributions:**

This paper studies the problem of finding Nash equilibrium in black-box games. The author first formulates the objective function as the summation of individual regrets. And then by first fitting the utility functions using Gaussian regression, the authors compute analytical maximum of the fitted function, and decide region of interest by incorporating upper confidence terms. Besides on these the next data point to be queried is determined. The authors also give theoretical analysis.

**Q2-3 Extent To Which Claims Are Supported By Evidence:**

2: Fair: the main claims are somewhat supported by evidence (but the experimental evaluation may be weak, or does not match entirely with the claims, important baselines may be missing, proofs contain important ideas but lack rigor, algorithmic details are only discussed superficially, references are imprecise, assumptions are not sufficiently motivated or explicated, etc.).

**Q2-4 Reproducibility:**

3: Good: key resources (e.g. proofs, code, data) are available and key details (e.g. proofs, experimental setup) are sufficiently well-described for competent researchers to confidently reproduce the main results.

**Q3 Main Strengths:**

Strength: problem is well-motivated.

**Q4 Main Weakness:**

The problem formulation is questionable. Missing discussion of important existing works. Lack of clarity of the methodologies.

**Q5 Detailed Comments To The Authors:**

While the problem of computing Nash in black-box games is well-motivated, it is hard to justify the approach taken by this paper. The main problem of this paper is that it lacks clarity of its approach and formulation. Some detailed comments:

1. First in the introduction, I believe putting centralized training for decentralized execution is misleading. CTDE normally refers to a multi-agent training procedure where agents at training time could access to the opponents' policy-wise information but are deciding action independently at evaluation time. For equilibrium computation in black-box games, it is more like taking a centralized game analyst's viewpoint.

2. About the objective function. I am not entirely sure what the no-regret learning objective aims to compute for. If the problem is to compute NE, then simply minimizing (1) suffice. For the no-regret learning objective are you viewing the time-averaging of x_t as the final solution? Is it going to be NE? In general-sum games a no-regret learning method typically converge to coarse correlated equilibrium. However your no-regret objective is still different from those as you are considering choosing profiles x in a centralized way. What kind of solution concept does it entail?

3. Equation (8) (9) are confusing, isn't (9) is likely to be small, because LCB is a kind of lower bound while UCB is an upper bound. What is the relation of (9) and the objective (1)?

4. Missing important previous work [1], which also considered using Gaussian Regression and UCB for learning Nash in black-box games. And (2) which also consider a black-box game setting and adopted a black-box algorithm for optimization.

[1] Learning Probably Approximately Correct Maximin Strategies in Simulation-Based Games with Infinite Strategy Spaces. AAMAS 2020, Marchesi et. al.

[2] Approximating Nash Equilibria in Normal-Form Games via Stochastic Optimization, ICLR 2024, Gemp et. al.

**Q9 Complying With Reviewing Instructions:**

Yes

---

> ### Author Rebuttal · Authors · 2024-04-02
>
> We thank the reviewer for the detailed feedback and for bringing our attention to a confusing point regarding the CTDE, we will revise our wording in that part to improve the clarity of our motivation. Below, we clarify the reviewer's concerns one by one. Due to a critical misunderstanding, we therefore respectfully request the reviewer to re-evaluate the merit of our paper.
>
> > About the objective function. I am not entirely sure what the no-regret learning objective aims to compute for. If the problem is to compute NE, then simply minimizing (1) suffice. For the no-regret learning objective are you viewing the time-averaging of x_t as the final solution? Is it going to be NE? In general-sum games a no-regret learning method typically converge to coarse correlated equilibrium. However your no-regret objective is still different from those as you are considering choosing profiles x in a centralized way. What kind of solution concept does it entail?
>
> We are somewhat confused by the reviewer's question. It seems possible that the reviewer may have made a typographical error, and that the reference to (1) should have been to (2) instead. We want to highlight that we are minimizing (2) instead of (1), since (1) is the definition of NE $x^*$.
>
> As for why are we viewing the time-averaging of $x_t$ as the final solution, we remark that our no *cumulative* regret learning guarantees minimizing (2) (i.e., the simple regret). Our results not only guarantee minimizing (2) but also guarantee the ability to approach Nash equilibrium and play a *high proportion of the learning period* (i.e., no (average)-regret). As for the difference from the convergence to coarse correlated equilibrium by the classical no-regret learning method, we have also discussed in our related work section that “these rules cause the empirical frequency distribution of play to converge to the set of correlated equilibria, which, while including Nash equilibria, is frequently much larger and not necessarily more desirable in terms of strategic outcomes.”
>
> > Equation (8) (9) are confusing, isn't (9) is likely to be small, because LCB is a kind of lower bound while UCB is an upper bound. What is the relation of (9) and the objective (1)?
>
> We want to highlight that (9) is based on (2) not on (1). Here (8) and (9) are the upper and lower confidence bounds of f(x) as defined in (2). It is expected that (9) is small. For example, since $f(x^*)=0$ means Nash Equilibrium is achieved at $x^*$, (9) is the lower confidence bound of $f$ should be smaller than 0 on $x^*$. Concretely, $LCB_{f,t}(x^*) < 0$. When $t$ goes to $\infty$, $LCB_{f,t}(x^*) \rightarrow 0$. Such property is also reflected in our theorem 1.
>
> > Missing important previous work [1], which also considered using Gaussian Regression and UCB for learning Nash in black-box games. And (2) which also consider a black-box game setting and adopted a black-box algorithm for optimization.
>
> Thank you for pointing out additional related works. For the AAMAS’20 paper, we remark that they focused on *two-player zero-sum* games while we focused on multi-player normal-form games. In addition, the analysis in the AAMAS’20 paper results relies on a suboptimal gap $\Delta^*$ on the denominator, resonating the multi-arm bandits' algorithm results before GP-UCB by Srinivas et al. 2009. The major problem with such results is that, as stated by Lattimore and Szepesvari 2020, $\Delta^*$ could be arbitrarily small and make the results unrealistic. At the same time, our results rely on recent advancements in Bayesian Optimization that use the maximum mutual information of GP in the regret bound, which is gap-independent.
>
> As for the ***recently published*** ICLR’24, we note that our work is complementary to the results presented in their paper, and can be viewed as an extension of the future work they proposed, as they themselves have mentioned in their publication: “ We consider bandit and vanilla SGD methods in this work, but these are only two of the many options now at our disposal (e.g, … Gaussian processes (Calandriello et al., 2022),...). Such approaches as well as generalizations of these techniques to extensive-form, imperfect-information games are promising directions for future work.”
>
> We will make sure to add a discussion of these related works in the future version of our paper.
>
> **Reference**:
>
> - Srinivas, N., Krause, A., Kakade, S. M., & Seeger, M. (2009). Gaussian process optimization in the bandit setting: No regret and experimental design. arXiv preprint arXiv:0912.3995.
>
> - Lattimore, Tor, and Csaba Szepesvári. Bandit algorithms. Cambridge University Press, 2020.

---

### Official Review · Reviewer_Axmo · 2024-03-23

**Q2-1 Originality-Novelty:** 3
**Q2-2 Correctness-Technical Quality:** 3
**Q2-5 Clarity Of Writing:** 4

**Q1 Summary And Contributions:**

The paper studies learning in a black box game with Bayesian optimization and shows that it arrives at equilibrium in sublinear time.

**Q2-3 Extent To Which Claims Are Supported By Evidence:**

3: Good: the main claims are supported by convincing evidence (in the form of adequate experimental evaluation, proofs, (pseudo-)code, references, assumptions).

**Q2-4 Reproducibility:**

3: Good: key resources (e.g. proofs, code, data) are available and key details (e.g. proofs, experimental setup) are sufficiently well-described for competent researchers to confidently reproduce the main results.

**Q3 Main Strengths:**

The use of Bayesian optimization in black-box games is very interesting, and the convergence rate result is very sound. The result is also high probability, which gives a very sound guarantee.

**Q4 Main Weakness:**

1. To my understanding, the existence of NE is not guaranteed in the general case. If NE does not exist, how should I interpretate the result?
2. I am not quite sure of the definition of no regret here. Corollary 2 shows sublinear regret with respect to f, which is the Nash equilibrium gap. This is different from the common definition of no-regret, which is defined with respect to a fixed sequence of adversarial opponents. Could you elaborate more on this?

**Q5 Detailed Comments To The Authors:**

1. Could you explain the results in games where Nash equilibrium does not exist?
2. Could you elaborate on the definition of no-regret?

**Q9 Complying With Reviewing Instructions:**

Yes

---

> ### Author Rebuttal · Authors · 2024-04-02
>
> We thank the reviewer for the encouraging feedback and for appreciating our results. Below, we clarify the reviewer's primary concerns, which we believe are due to misunderstandings about our model. Therefore, we respectfully request the reviewer to re-evaluate the merit of our paper.
>
> > Could you explain the results in games where Nash equilibrium does not exist?
>
> Thanks for pointing out a confusing point. Our paper specifically focuses on finite games, which involve a finite number of players and a finite number of actions for each player. It is well-established, as demonstrated by John Nash in his 1950 paper "Non-cooperative games," that every finite game possesses at least one Nash equilibrium, commonly known as the "Nash existence theorem." We will make sure to emphasize this clarification in the future version of our paper. It is unclear whether the reviewer was referring to the application of our results to infinite games. However, neither our results nor previous studies have explored this context, and we believe it falls outside the scope of our current research. Nevertheless, our current findings may serve as a source of inspiration for developing approximate solutions for Nash equilibria in infinite games.
>
> > Could you elaborate on the definition of no-regret?
>
> We would like to clarify that our definition of no-regret is not different from the classical definition of no-regret when treating $-f(x)$ as the reward function to be maximized. As we defined in Section3, “We define $f(x^t) − f (x^∗) = f (x^t)$ as regret, since $f (x^∗) = 0$ for NE”, where the regret is indeed defined with respect to a fixed sequence of $x^*$. As we have introduced in equation (2), the value f(x) defines the sum over all agents’ utility gain if they deviate from the given strategy profile x, therefore, a no-regret algorithm according to our definition $\lim_{T \rightarrow \infty} \frac{1}{T} \sum_{t \in [T]} f(x^t)$ not only guarantees the convergence to NE (i.e., the simple regret), but also guarantees the that the NE is played a *high proportion of the learning period* (i.e., leading to asymptotic no-average-regret).  In summary, when treating $-f(x)$ as the reward function, and, $-f(x^*)=0$ as the optimal reward, we follow the conventional definition of no-regret, e.g., in Srinivas, Niranjan, et al. 2009.
>
>
> **Reference**:
> Srinivas, Niranjan, et al. "Gaussian process optimization in the bandit setting: No regret and experimental design." ICML 2010.

---

### Meta-Review · Area_Chair_CY78 · 2024-04-15

This paper proposes an approach based on Gaussian processes for learning Nash equilibria in games, using only queries of a black-box utility function. It then provides no-regret theoretical guarantees, and is accompanied by experimental results. The paper is well-written, and technically sound, and the experimental results are reproducible and sufficient. There was some confusion regarding the setup, and concerns regarding the generalizability of the results, and positioning in the literature (given some missing literature some reviewers brought up). I encourage the authors to incorporate the feedback in preparing the next version of the paper.